# Fast Vortex Particle Method for Fluid-Character Interaction

Asger Meldgaard*
University of Copenhagen

Kenny Erleben†
University of Copenhagen

Sune Darkner‡
University of Copenhagen

## ABSTRACT

High fidelity interactions between game characters and gaseous effects like smoke, fire and explosions are often neglected in real-time applications due to the high computational cost of simulating fluids. In addition, the pose of game characters is only known at run-time as it depends on input from the user. Thus simulation-suitable representations of surface geometry must be generated on the fly. Common approaches like conversion into signed distance fields are not feasible for high-resolution geometry due to the computational cost and the amount of memory required on the GPU to store these fields. We present a purely vortex particle based fluid model for games which is capable of resolving the collision between fluids and complex objects such as moving game characters in real time. To handle collisions, we use a collocation method which only require a set of disassociated particles stuck to collision surfaces. Contrary to most other vorticity based methods, we use a simple inversion free approach to obtain the collision velocity field on surfaces while at the same time avoiding the expensive pressure projection step associated with pressure based fluid solvers.

**Index Terms:** Game physics—Simulation—Fluids—Real-time graphics; Animation—Visualization—Game characters—Vortex method

## 1 INTRODUCTION

Real-time simulation of fluids is a valuable addition to interactive applications such as games or virtual reality. While smoke, fire and explosions are key components in immersive gaming experiences, physically correct simulation of fluids is usually not feasible at high-resolution. Smoke and explosions are sometimes pre-simulated in high-resolution and played back in real-time but this approach precludes any interactions with characters which depends on the input from the user and can only be known at run-time. To represent interactions with fluids in a believable way, a detailed representation of the fluid velocity field close to the character surface is needed to match the high-resolution geometric models used for characters in modern games. For game applications, vortex particle methods provide an interesting alternative to the more common velocity-pressure based representation of fluid state where the iterative pressure projection step constitutes a computational bottleneck for real-time applications.

By evolving a vorticity field discretized by particles, no pressure projection is needed and the divergence free velocity field can be derived by using the Biot-Savart law. In addition, vortex methods generate unbounded continuous solutions. It also allows for easy adaptation to the different applications for which fluid simulations may be required in games. Another use case besides smoke and fire, is wind systems. Our proposed method is useful here as well since the computational cost scales

*e-mail: asger.meldgaard@di.ku.dk
†e-mail: kenny@di.ku.dk
‡e-mail: darkner@di.ku.dk

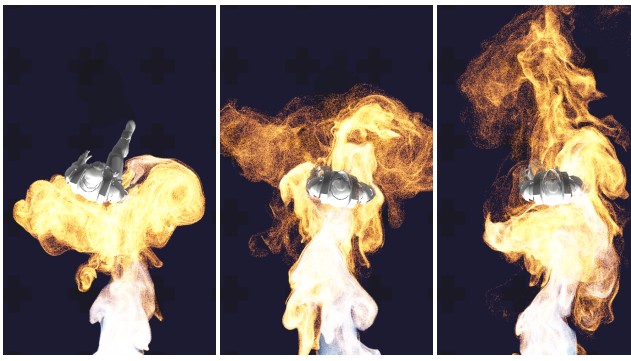

Figure 1: Our vortex method simulates the interaction of a moving character and fluids in real-time. A typical game effect is shown of a moving game character being hit by a plume of fire.

with the number of velocity evaluations. Thus a few leaves being moved in a wind field does not require the calculation of the fluid velocity field throughout the domain which would be required for prevalent methods based on pressure projection. Pure vortex methods have been used for free-surface liquid simulation [4] although such methods are generally not faster than their counterparts based on pressure projection. In this work, we limit fluid simulation to mean gaseous fluids for which vortex methods are highly suitable, while other methods are better suited for water which is of course an equally important component in virtual worlds.

Vortex methods allow for detailed real-time simulation in the absence of kinematic boundary conditions, but it is not possible to define the boundary conditions in terms of vorticity. Boundary element methods are common approaches for enforcing kinematic boundary conditions in vorticity based fluid methods but they are computationally expensive and the methods used in most vortex codes like [24] and [6] to obtain a vortex sheet strength across collision surfaces are not very suitable for real-time applications as they require surface integration and inversion of a coupling matrix. Instead, we use a collocation method inspired by the method of [5]. This cheap collision method allows for believable transfer of complex character motion into the surrounding fluid. The influences of moving surfaces are taken into account by scattering source points on collision surfaces and tuning their attributes based on the relative velocity of the surface and the surrounding fluid. As a surface only method, it is decoupled from the underlying high-resolution geometry and only requires source points scattered across the surface with an area attribute and a normal. In our method, these source points are attached to the character rig and moved with the underlying surface. We use the continuous velocity field of the vortex particles to adaptively sample the fluid velocity field on the surface at the required resolution. Therefore the method is effectively decoupled from the underlying geometry and independent of any collision detection steps on the high-resolution mesh.

To further enhance realism, we account for additional rotation

introduced in the thin viscous boundary layer at the collision interface by initialization new vortex particles where the tangential velocity of the fluid tends to zero due to shear stresses [2]. The continuous velocity fields generated by the vortex particles allow many different kinds of fluid representations. Here we choose to visualize the effect with large numbers of passive tracer particles advected in the ambient velocity field. To make this feasible, we first evaluate the velocity field on an intermediate scratchpad grid and then broadcast the field to the tracer particles using linear interpolation. Some examples of our method are shown in Figure 1 where a moving game character is being hit by a plume. Figure 2, shows a similar effect where turbulence is generated solely at the collision interface leading to intricate swirly fluid motion downstream. With small altercations to the method, and the introduction of a density attribute on the scratchpad grid, it is also possible to represent fluids as volumetric effects. An extra ray-marching step in then required in the pixel shader to render the effect but there is no need for millions of velocity interpolations which is the most expensive part of the particle based fluid representation.

The time allocated to physics update in most AAA games is limited, thus a real-time capable fluid solver is not necessarily efficient enough for a game. Feasible physics update times are typically in the order of $\sim 500\ \mu s$ for effects used in many places throughout the scene. Our method allows high-resolution simulation times of approximately $1-2\ ms$. Thus, our method is viable though should be used sparingly.

Most time is spent interpolating and advecting the velocity field on the tracer particles where we typically use several millions to render the effect as a continuous looking fluid. Rendering millions of particles is the most time consuming part. For the examples shown here, we use $2.7 \cdot 10^6$ million tracer particles unless states otherwise and the entire game simulation runs at $\sim 250\ FPS$ on a gaming laptop with a RTX2080 Max-Q GPU.

The main contributions in this work are:

- A purely vortex based particle method for games. Our method simulates intricate turbulence generation without the need for any pressure projection steps.

- The method does not rely on an underlying regular data structure and creates continuous velocity fields. The fields can be sampled adaptively depending on the computational budget. Fluids can be represented as particles, in volumes or in textures without affecting the underlying dynamics.

- We adopt a simple approach for handling complex boundaries and infer an appropriate image vorticity field across surfaces without the need for matrix inversion steps.

- A computationally light-weight density representation based on tracer particles which are advected passively in the velocity field and otherwise decoupled from the physics update step.

## 2  RELATED WORK

Fluid simulation in graphics is a vast field having been developed over the years to facilitate increased fidelity and efficiency. Textbooks like [9] or [29] give a good introduction to many of these contributions. In this work, we primarily focuses on the subset of methods specifically intended for real-time applications. Simulation techniques using regular grids as the underlying discretization of space are widely based on the work of [28]. These methods lend themselves particularly well to GPU architectures. Particle based methods like those developed by [18] and [31] have also been used

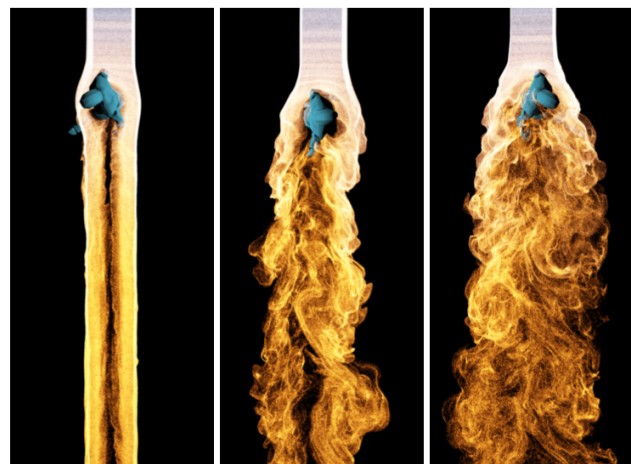

Figure 2: A stream of tracer particles is emitted from the top. The stream hits an animating character. Vortex particles are initialized in the collision boundary layer. The amount of vortex shedding is increased from left to right to achieve different looks.

to simulate fluids in real time while simulations on unstructured grids like [20] and surface-only methods like those used by [13] are particularly useful for accurate handling of interfaces and collisions. Sparse data structures like the tall-grid cells used by [11], octrees employed by [1] and VDB volumes [21] have accelerated fluid simulation even further by limiting the high-resolution simulation domain to areas where it is needed such as at a free surface or near obstacles. The above mentioned methods facilitate real-time fluid simulation but current methods must be accelerated even further to allow for practical use in large game worlds where fluid simulation can only occupy a fraction of the available computational budget.

### 2.1  Vortex sheets, particles and filaments

Vortex methods commonly use particles, filaments and sheets to discretize the vorticity field from which a divergence free velocity field can be derived. While the state of a fluid can be evolved solely on the basis of a vorticity field, vortex particles have also been used to enrich low-resolution fluid simulations as shown by [27]. [23] and [25] used vortex particles for procedural turbulence advection in an underlying velocity based fluid solver. Vortex particles provide a cheap way of introducing details in a fluid simulation, and they are good candidates for game applications where particle systems are used frequently. However, the kinematic boundary conditions required on collision surfaces are not trivial to enforce. Pure vortex particle methods like those described by [24] require inversion of a pseudo-inverse collision matrix for each time-step making this approach unsuitable for real-time applications. Evaluation of the velocity field requires querying all vortex particles in the simulation whereas our proposed method simplifies this by using a clamped vortex kernel. Hybrid grid and particle methods like [33] have been proposed which are efficient for fast computation of long-range interactions on an underlying grid, while using particle-particle interactions to model short-range interactions. Methods such as the fast multipole method by [15] can accelerate the expensive summation from $\vec{O}(NM)$ to $\vec{O}(N+M)$ for N vortex particles and M locations. The FMM adds a large computational overhead and only outperforms direct summation for large particle counts where $N > 100000$ [33] which is far above the vortex particle counts used in this work.

Vortex sheets are boundary element methods used to evolve the vorticity field on interfaces such as the interface between collision

objects and fluids, and the interfaces between fluids. [10] used a vortex sheet method to model smoke plumes in linear time using the fast multipole method. [26] simulated smoke plumes by combining a vortex sheet method with an Eulerian grid solver to handle collisions. Closely related methods like [13] were applied for surface only liquids. Vortex filament methods use closed loops of vorticity to represent the state of fluids. This approach is suitable for gaseous plumes. [32], [30] and [7] demonstrated how these methods provide a very cheap way of creating the intricate dynamics of smoke while [30] used vortex filaments and filament shredding around solid objects to handle collisions with objects. Vortex shredding entails seeding of vorticity in the collision boundary layer, and we adopt this approach to enrich fluid collisions with characters. Vortex filaments and sheets represent a vorticity field that is divergence free by default, and vortex stretching is handled trivially. Unfortunately, both methods require re-discretization as simulations evolve in order to insure fidelity. This also entails that the number of vortex elements may grow beyond what is feasible for real-time scenarios.

## 2.2 Fast particle based fluid approaches

Particle-based physics solvers are used abundantly in games. [19] presented a unified particle based framework for real-time physics based on position based dynamics (PBD). PBD is also applicable to fluid simulation as shown by [18]. While PBD is a generalizeable and robust simulation method, it can be expensive for detailed simulations where large quantities of particles are required. In addition, sufficient iterations are required to ensure incompressible velocity fields and stable simulations. Vortex methods are difficult to extend beyond their applications for gaseous fluid phenomena whereas PBD trivially handles free surfaces. On the other hand, a vortex particle codes only need to advect a small number of vortex particles to generate complex turbulence patterns. The same level of detail is generally not feasible for PBD with the current computational budget of games.

## 2.3 Fast sparse grid-based approaches

The regular data structures used in grid-based fluid solvers lend themselves well to GPU implementations. Fluid solvers based on the original work by [28] like [16] discretizes collision objects on the simulation grid which entails that increased collision fidelity requires a global refinement of the simulation domain. [8] proposed a variational frame-work to address this issue while the advent of sparse data structures like tall cells by [11], or octrees by [3], and recently GPU-optimized VDB volumes by [22] allows for a local discretizations in the vicinity of free surfaces and collision objects. While in particular nanoVDB's presented by [22] can simulate fluids in unprecedented detail, they still require a volumetric representation of collision geometry. For the deforming character surface, this requires access to the character mesh at run-time and an update of the VDB data-structure. Our method uses a surface-only approach and only needs to update the position and orientations of the discrete source points scattered across the mesh.

## 3  METHOD

In-compressible fluids are governed by the mass conservation relation: $\nabla \cdot \vec{v}$ and the Navier-Stokes equation for conservation of momentum:

$$\frac{\partial \vec{v}(\vec{x},t)}{\partial t} + (\vec{v}\cdot\nabla)\,\vec{v} = \frac{1}{\rho}\left(\vec{f}+\mu\nabla^2\vec{v}-\nabla p\right) \qquad (1)$$

where $p$ is the pressure, $\rho$ is the density, which is assumed to be constant, and $\vec{f}$ are external forces like gravity and baroclinity and $\mu$ is the dynamic viscosity. It is possible to define an alternative

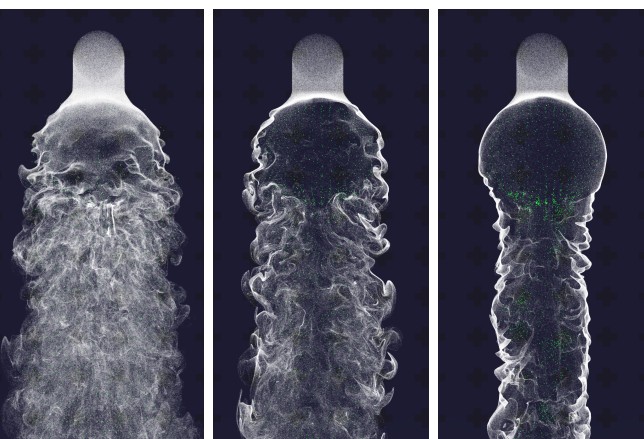

Figure 3: By measuring the free-space velocity field at collocation points on geometry surfaces, we create an accurate divergence free collision velocity. Here, a stream of particles moves over a simple sphere. The accuracy of the collision field is dependent on the collocation point density. We found that the collision method can generate accurate collision fields as demonstrated on the simple sphere. To control the look of the smoke, We distribute vortex particles on the surface and shred them into the ambient flow.

version of the momentum equations based on vorticity, the vector field describing the rotation of fluid:

$$\vec{\omega} = \nabla \times \vec{v} \qquad (2)$$

Taking the curl of the momentum equation 1 yields a new equation for the time evolution of vorticity:

$$\frac{\partial \vec{\omega}(\vec{x},t)}{\partial t} + (\vec{v}\cdot\nabla)\,\vec{\omega} + (\vec{\omega}\cdot\nabla)\,\vec{v} = \frac{1}{\rho}\left(\nabla \times \vec{f} + \mu\nabla^2\omega\right). \qquad (3)$$

The vorticity field can be evolved in time according to equation 3 without the need for any pressure projection steps. A fractional step method is commonly employed for time integration. In the fractional step method, the vorticity field is stepped in time assuming no kinematic boundary conditions. The initial free-space solution is then corrected for collisions by submerging collision objects into the initial unperturbed field. The relative velocity between the objects and the ambient fluids allows us to calculate a superimposed collision field from objects in the scene such that the normal component of velocity is minimized at the surface. Vortex particle methods are Lagrangian in nature and handle advection of vorticity $((\vec{v}\cdot\nabla)\vec{\omega})$ trivially by storing vorticity on entities that move in the ambient flow. The third term on the left $((\vec{\omega}\cdot\nabla)\vec{v})$ accounts for vortex stretching and is a feature of 3D fluids only since the vorticity vector is always perpendicular to the velocity field for 2D fluids. Vortex stretching transfers large scale rotation into smaller vorticles. In vortex particle methods, vortex stretching requires special attention (an estimate of $\nabla\vec{v}$) and diffusion is required to insure stability. For game applications, stretch and diffusion could be neglected without very noticeable impacts on realism but we include these terms for completeness. Vortex stretch degrades performance since the method we employ requires two evaluations of the velocity field for each particle and requires vortex diffusion to insure that the vorticity field stays approximately divergence free.

## 3.1 Fundamental solutions

A well behaved divergence free velocity field $v(\vec{x},t)$ can be represented through a vector potential $\vec{A}$. The velocity field is obtained by

taking the curl of $\vec{A}$ which ensures that the divergence of the velocity field is zero,

$$\vec{v}(\vec{x},t) = \nabla \times \vec{A}(\vec{x},t). \tag{4}$$

$\vec{A}$ is degenerate since the addition of any curl-free vector field yields the same velocity. To tie down the vector potential, it can be assumed that $\nabla \cdot \vec{A}(\vec{x},t) = 0$. In that case the vorticity field $\omega(\vec{x},t)$ and the vector potential are related through a vector Laplacian,

$$\vec{\omega}(\vec{x},t) = \nabla \times \nabla \times \Psi(\vec{x},t) = -\nabla^2 \Psi(\vec{x},t). \tag{5}$$

In the absence of boundaries and under the assumption that the velocity field goes to zero at infinity, the solution can be composed of a linear combination of fundamental free-space solutions or Green's functions. The vector potential is obtained by integrating the fundamental solutions over the domain,

$$\vec{A}(\vec{x},t) = \frac{1}{4\pi} \int_V \frac{\vec{\omega}(\vec{x}',t)}{\|\vec{x}-\vec{x}'\|} d\vec{x}'. \tag{6}$$

Taking the curl of equation 6 leads to the *Biot-Savart* formula for the velocity field,

$$\vec{v}(\vec{x},t) = \frac{1}{4\pi} \int_V \vec{\omega}(\vec{x}',t) \times \frac{\vec{x}-\vec{x}'}{\|\vec{x}-\vec{x}'\|^3} d\vec{x}'. \tag{7}$$

The vortex particles act as quadrature points in the discrete version of equation 7. To avoid singularities when $\vec{x} = \vec{x}'$, we use a mollified solution similar to [12]. This is analogous to the inclusion of a smoothing radius $h$ in the denominator which effectively limits the minimum swirl size,

$$\vec{v}(\vec{x},t) = \frac{1}{4\pi} \sum_i V_i \omega_i \times \frac{\vec{x}-\vec{x}'_i}{\left(h^2 + \|\vec{x}-\vec{x}'_i\|^2\right)^{\frac{3}{2}}}. \tag{8}$$

$$\vec{u}_p(\vec{x},t) = \frac{1}{4\pi} \sum_i V_i \omega_i \times \frac{\vec{x}-\vec{x}'_i}{\left(h^2 + \|\vec{x}-\vec{x}'_i\|^2\right)^{\frac{3}{2}}}. \tag{9}$$

The vortex blob volume is given by $\vec{V}_i$, $\omega_i$ is the blob vortex density, and $\vec{w}_i = V_i \omega_i$ is the vorticity stored on each vortex particle. Equation 9 is used to obtain the velocity field anywhere in space. To avoid iterating over every vortex particle in the simulation, we use a nearest neighbour search based on [17] to only query the nearest particles which is sufficient for game applications although physical accuracy would require the contribution from all particles in the simulation either through direct summation or multipole methods. We optimize the simulation further by excluding vortex particles from the simulation when their vorticity falls below a certain threshold. These particles are then recycled, either by emission from sources, or they are re-positioned close to surfaces where their vorticity attribute gets reinitialized.

The time dependent evolution of the vorticity is driven by vortex advection, stretching and diffusion. It is possible to get believable fluid-like motion by only using vorticity advection but vortex stretching can easily be included with the vortex segment approach introduced by [33]. The stretching term in equation 3, is a measure of the velocity gradient in the direction of the vorticity vector scaled by the vorticity magnitude. Vortex particles do not have a spatial extent but we can measure the gradient of the velocity field in the direction of the vorticity vector by converting each particle into a small vortex segment as shown in figure 5 and measuring the gradient over the segment. The vorticity vector is then updated as:

$$\vec{w} \leftarrow \vec{w} + \|w\| \frac{\Delta t}{h} \left(\vec{v}(\vec{q}_1) - \vec{v}(\vec{q}_0)\right) \tag{10}$$

where $\vec{q}_1$ and $\vec{q}_0$ are the positions of the vortex segment ends, $\vec{q}_0 = \vec{x} + \frac{h}{2}$ and $\vec{q}_1 = \vec{x} - \frac{h}{2}$. Vortex stretching converts large swirls into smaller swirls and this can eventually lead to instabilities if the process is allowed to proceed unimpeded. Vortex diffusion is required to ensure stability when the vorticity field is undergoing stretch. The particle strength exchange method gradually homogenizes the vorticity field and insures that it remains nearly divergence free. Therefore, the diffusion $d\omega/dt = \mu \nabla^2 \omega$ is approximated with,

$$\omega \leftarrow \omega + \Delta t \frac{2\nu}{\sigma} \sum_q \left(\vec{V}_q \omega_q - V\omega\right) \zeta(\vec{x},\vec{x}'_q) \tag{11}$$

where $\zeta$ is a normalized Gaussian,

$$\zeta(\vec{x},\vec{x}') = \frac{1}{\sigma^3(2\pi)^{3/2}} e^{-\frac{\vec{x}-\vec{x}'^2}{2\sigma^2}}, \tag{12}$$

and where the viscosity $\nu$ and the smoothing radius $\sigma$ are exposed parameters.

## 3.2 Boundary conditions

To enforce boundary conditions, source points are stuck to collision surface as illustrated in Figure 4. The ambient velocity field is measured at the source points and we use them to generate an image velocity field which minimizes the normal component of flow. Figure 4 shows the configuration of 512 source points on the surface of a character. The ambient velocity field consists of $\vec{u}^s$ which is the velocity of the surface itself, $\vec{u}^\infty$ is a superimposed harmonic velocity (such as an initial flow velocity) and $\vec{u}^p$ is the turbulent velocity generated by the vortex particles in the ambient fluid.

We could view the points on the collision surface as vortex particles with a collision vorticity which is indeed a common procedure in vortex particle methods. An optimal vortex sheet strength can be posed as a regression problem.

$$\gamma_j^* = \arg\min_{\gamma_j} \left\{ \left(\vec{u}_i^s - \vec{u}_i^\infty - \vec{u}_i^p + \vec{v}(\vec{x}_i,\gamma_j)\right) \cdot \vec{n}_i \right\}. \tag{13}$$

Here $\gamma^*$ is the optimal vortex sheet strength, $\vec{u}_i^s$, $\vec{u}_i^\infty$ and $\vec{u}_i^p$ are the ambient velocity components measured at collocation points at the surface position $x_i$ and $\vec{v}(\vec{x}_i,\gamma_j)$ is the image velocity field,

$$\vec{v}(\vec{x},\gamma_j) = \frac{1}{4\pi} \sum_j A_j \gamma_j \times \frac{\vec{x}-\vec{x}'_j}{\left(h^2 + \|\vec{x}-\vec{x}'_j\|^2\right)^{\frac{3}{2}}}. \tag{14}$$

The index $j$ denotes vortex source points. Each source point stores a tangential vortex vector and the generated velocity field is measured at point $x_i$. We require that $i > 2j$ to have an over-determined system of equations which ensures a unique solution.

Unfortunately, this method is not very suitable for real-time applications since it requires the pseudo inverse. Solving the linear system of equations to resolve collisions becomes a significant computational bottleneck.

[5] introduced an alternative for simply connected closed surfaces where the matrix inversion step is mitigated. This approach is ideal for game applications. We find the optimal Rankine collision field $\vec{v}_R$ by treating each source point as a Rankine field source. Then we obtain the Rankine image field by adding the contributions from the source points,

$$\vec{v}_R(\vec{x}) = \int_S \vec{n} \cdot \left(\vec{u}_s - \left(\vec{u}_p + \vec{u}_\infty\right)\right) \nabla G \, d\vec{x}. \tag{15}$$

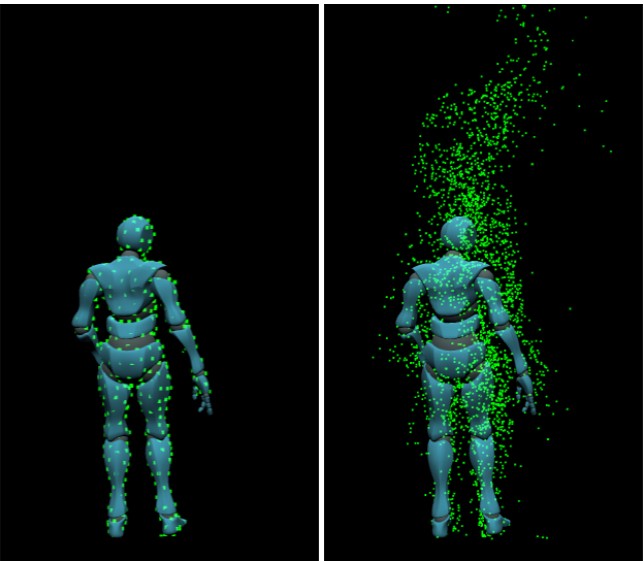

Figure 4: Source points on collision surfaces (left). The points are moved by the underlying rig without dependence on the surface mesh. We measure the relative velocity of the fluid at the position of the points and calculate the appropriate source strengths to enforce the boundary conditions. Additional vortex particles (left) are advected in the ambient velocity field.

Here $G$ is the Rankine Green's function $1/\|\vec{x} - \vec{x}'\|$ and $\vec{v}_R(\vec{x})$ is the Rankine image velocity field over the surface. By superimposing this field on the free-space solution, the normal flow is minimized as shown in detail by [5]. While this approach only works for each collision surface in isolation, the continual measurement of the surface velocity field on all surfaces ensures that the perturbations to the velocity field created by one object is mapped onto the surface of all other objects in the scene. To carry out this integral, we discretize it by using the mollified Green's function and discrete integration over the source points on the collision surface,

$$\vec{v}_R(\vec{x}) = \sum_j A_j \vec{n}_j \cdot (\vec{u}_s(\vec{x}_j) - \vec{u}_p(\vec{x}_j) - \vec{u}_\infty(\vec{x}_j)) \nabla G(\vec{x} - \vec{x}_j). \quad (16)$$

The Rankine collision field is divergence free but the interesting fluid behaviour associated with surface interactions requires an estimate of the fluid rotation generated in the thin viscous boundary layer that forms in real fluids. We use a simple model to transfer rotation to vortex particles close to the surface. The procedure is shown in Figure 6. The tangential component is extracted from $\vec{v}_R$. The initialized vortex vector is perpendicular to the surface normal and the tangential velocity component. Vortex particles can be initialized with a small normal displacement $\varepsilon$ from the surface. The vorticity is determined such that the generated velocity field cancels the tangential part of $v_R$ at the surface. The vortex vector is stored on the surface particles and mapped to nearby free-flowing vortex particles with an exponentially decreasing kernel to simulate turbulence generation in the boundary layer.

### 3.3 Velocity Broadcast

It is possible to create appealing turbulent fluid motion with a small number of vortex particles but we still need a way of representing density. To render the effect, we distinguish between vortex particles and tracer particles. The velocity field is calculated directly on each vortex particle but this is not feasible for the millions of tracer particles used to render the effect. To update the velocity on millions of tracer particles in real time, we found that the best solution is an

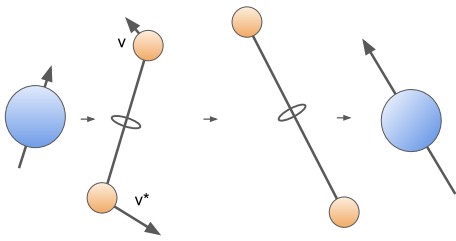

Figure 5: A vortex particle is converted to a vortex segment. The velocity is evaluated at each end of the segment. When the segment has been stretched in the velocity field it is converted back into a vortex particle.

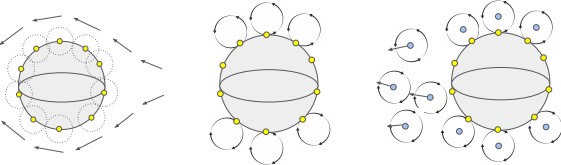

Figure 6: The relative velocity between the surface and the ambient fluid is measured and the no-through Rankine collision field is calculated using equation 16 (left). We initialize vorticity on the surface to match the tangential component of the Rankine velocity (middle), and map vorticity to the particles in the surrounding flow (right).

underlying scratchpad grid. The velocity field is calculated on grid nodes and tri-linear interpolation is used to update the velocity of the tracer particles within the grid. A second-order Adams-Bashforth scheme was used for time integration. We found that this explicit integration method produces well-defined turbulence patterns while only requiring storage of the velocity of the previous time step.

$$\vec{x}_t^{(n+1)} = \vec{x}_t^n + \left( \frac{3}{2} \vec{v}_t^n - \frac{1}{2} \vec{v}_t^{(n-1)} \right). \quad (17)$$

The grid is sparse in the sense that each voxel only query the nearest vortex particles and empty grid regions are cheap to update. It is possible to use large grid domains without a significant compromise to efficiency. Tracer particles simply trace the velocity field, but they are not required to update the dynamics. Since each tracer particle only needs to source the velocity from the grid, millions of particles can be traced in real time though sub-millisecond performance restricts the count to $\sim 10^6$ on a RTX2080 Max-Q GPU. For all the simulations shown in this work, we use 4000-8000 vortex particles.

### 4 RESULTS

We have simulated several examples showing the interaction between streams of fluids and different game characters. The motion capture library and character geometries from `mixamo.com` were used in the simulations. This library contains complex animations like dancing, running and jumping. Different characters with significant shape variations are used to further demonstrate the

Table 1: Time measurements of the physics update with different combinations of tracer particles and vortex particles. The number in brackets denotes the upper limit of particles queries admitted for each velocity evaluation. With $\infty$, we denote an unlimited number of queries. Unless explicitly mentioned, the illustrations presented in this work used $140^3$ tracer particles, $16^3$ vortex particles, $100^3$ grid nodes and $32$ as the query limit. All steps of our algorithm are implemented on the GPU and only the configuration of the character rig (bone transforms) needs to be transferred on each time step. We list this step separately as the access to the bone transforms will be readily available on the GPU for most game applications. We use a RTX2080 Max-Q found in high end gaming laptops for the simulations.

| Name | Tracer pts / Vortex pts / Grid | Time ($ms$) |
|---|---|---|
| *Rig Transfer* | - | 0.6 |
| **Laminar Beam** | $100^3$ / $16^3[64]$ / $100^3$ | $<0.3$ |
| **Laminar Beam** | $100^3$ / $20^3[64]$ / $100^3$ | 0.4 |
| **Laminar Beam** | $140^3$ / $20^3[64]$ / $100^3$ | 3.1 |
| **Laminar Beam** | $140^3$ / $20^3[32]$ / $100^3$ | 2.3 |
| **Buoyancy Driven** | $100^3$ / $32^3[32]$ / $100^3$ | 2.1 |
| **Buoyancy Driven** | $140^3$ / $32^3[32]$ / $100^3$ | 5.1 |
| **Buoyancy Driven** | $140^3$ / $16^3(\infty)$ / $100^3$ | 21.1 |

versatility of the method. With just 4096 vortex particles, our method can simulate fluids with rich dynamics. The physics easily fits within the computational budget of most games and we are able to update millions of tracer particles in real time. Tracing the velocity field constitutes the computational bottleneck of our method. We found that tracer particle counts up to $\sim 10^6$ are feasible for sub-millisecond performance.

Figure 1 and 2 depicts streams of fluid hitting characters in motion. The turbulence generation from the boundary layer is evident in Figure 2 and our method can also be used to simulate a variety of flame-like effects by seeding vortex particles with random vorticities at the fluid source as shown in Figure 1. Figure 3 shows collisions with a simple sphere object. The source particles on the collision surface creates accurate collision field though it requires a sufficiently large search radius. The source particles on surfaces are treated like the vortex particles in the surrounding fluid and it is important to include a sufficient number of source particles to resolve the collisions accurately. The side-by-side simulations in Figure 7 shows two similar scenarios with different numbers of tracer particles. Including rendering, we can simulate more than 800 fps with $10^6$ tracer particles and the dynamics are unchanged by the tracer particle count.

## 5 DISCUSSION AND LIMITATIONS

Pure vortex particle methods are well suited for real-time fluid simulation in game applications but have not been used widely. The method outlined here is simple to implement and fast enough to fit within the computational budget of most games. In this work, we have explored a limited set of applications, specifically, the interactions between characters and fluids which is a particular challenge with existing methods. By placing vortex particles on surfaces and using the matrix-free collision method, this can be handled easily with the proposed method. Several improvements are possible. The placement of source particles on collision geometry is fixed at runtime yet it could be advantageous to place them dynamically in the vicinity of fluid density. In particular, this may be required for large game worlds where all surfaces could potentially be collision surfaces. The continuous velocity fields created by the vortex particles is a decidedly advantageous feature of pure vortex methods. Since the evaluation of the velocity field represents the computational bottleneck, the number of velocity samples can be adapted to fit the

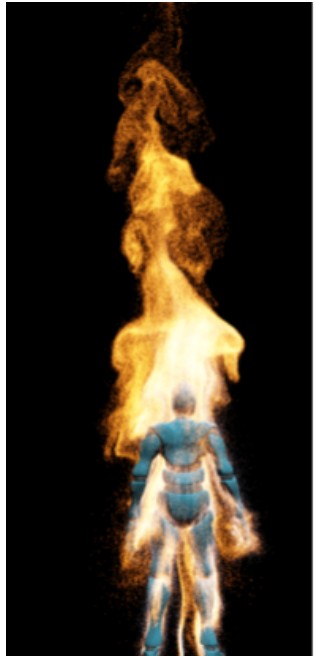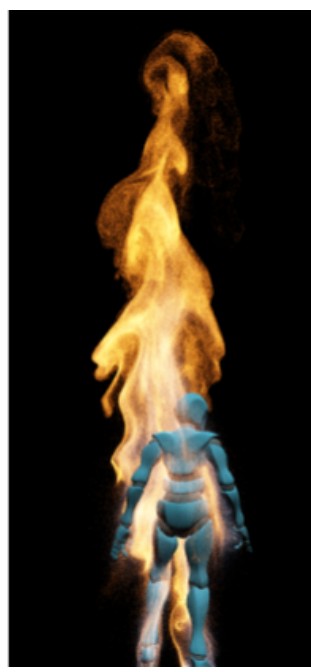

Figure 7: 4096 vortex particles instigate detailed fluid motion in a field of tracer particles. The detail of the dynamics are unchanged by the number of tracer particles but to render the effect as a seemingly continuous fluid, large numbers of tracer particles are needed. Here we show a comparison between $10^6$ tracer particles used in the left image and $2.74 * 10^6$ tracer particles are used on the right. The left simulation including rendering runs at $\sim 900$ fps and the right simulation runs at $\sim 300$ fps. Thus future work or production may substitute the millions of particles for fewer numbers of a more suitable tracer entity such as animated sprites, sparks or volumes to significantly speed up simulation.

computational budget and the level of detail needed for a particular application. A continuous velocity field entails that it is easy to swap the tracer particles for other density representations such as grid-based density fields or texture representations. Vortex methods are well suited for gaseous fluids but they are difficult to adapt to other fluid phenomena. In particular, the inclusion of a free surface required for liquids is not straight-forward although approaches such as [14] is an example of a vortex method for liquids. These approaches are not necessarily more efficient that their velocity based counterparts.

## 6 CONCLUSION

We have presented a vortex based fluid solver capable of handling the intricate collisions between fluids and game characters. It is the first method to specifically target these interactions and resolve them at a high level of detail. Our method is fast enough for practical use in interactive applications like games or VR and represents an additional step towards bringing realism to fluid simulations for these kinds of applications.

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
