# OpenReview forum: "Fast Vortex Particle Method for Fluid-Character Interaction"
_graphicsinterface.org/Graphics_Interface/2022/Conference — GI 2022_

### Official Review · Reviewer_XJhg · 2022-04-13
**A useful technique for games**

**Rating:** 8
**Confidence:** 2

**Review:**

Fluid animation is not my expertise so my review only reflects my best effort to understand the paper. I do research on character animation, so I guess that's why this paper is assigned to me for review. But this paper is a pure fluid animation paper, although it has the word "character" in its title.

The paper uses a vortex particle method for simulating fluid-character interaction for interactive game applications. The vortex particles can then generate continuous velocity fields that can then advect tracer particles for visualization. Boundary conditions are handled by a collocation method that avoids expensive matrix inversions.

I don't usually follow the fluid animation literature, so I can't say for sure how significant the contributions are with respect to prior art. But the components described in the paper all sound reasonable, and the results are fast and look believable. As always, I think faster-than-real-time techniques are valuable. The paper is also well written, with a few typos here and there.

---

### Official Review · Reviewer_ayz3 · 2022-04-13
**Innovative system for a very fast simulation of gaseous fluids interacting with colliding objects**

**Rating:** 7
**Confidence:** 4

**Review:**

This paper perfect for video game applications. The proposed approach uses vortex particles for the simulation. The strategy assembles clever ways to do the computation: no need for matrix inversion to get the velocity, no pressure projection, adaptive sampling of the velocity field (can adapt to computation budget), and collocation method. The approach has the following advantages: divergence-free velocity field. The results are very promising

The paper is well structured and written (despite some typos here and there, see below).

While the approach provides good results and seems to be well designed, the paper lacks some comparison.

Seems hard to control from a user perspective as the behavior seems to vary according to:
* Tracer particle count
* Source particle count
* Vorticity particle count
Also, from a simulation standpoint, if the result significantly varies with respect to the number of tracer particle, source particle, and vorticity particle, it is not clear if the approach converges to a stable result. I would like to see if the result becomes more and more similar with increasing number of tracer, source, and vorticity particles.

As the number of particles influences the result, it would be important to provide the reader with an intuition about how to adjust the numbers for each type of particle to get a specific result.

Another question related to the above: how does the result change when increasing/decreasing the velocity grid size?

How does the approach compare to an offline and high-accuracy simulation?

Test points, source particles, source points, and collocation points: Not clear in the paper if these are the same or how they are related to each other.

It was not clear to me if the tracer particles are seeded only at the source of gaseous fluid or if some are reseeded/redistributed throughout the simulation domain (like in FLIP for example).

I could not find an explicit explanation of how the gas is rendered. Is it some transparent blob around each tracer particle?

Fig. 3, what is the difference left to right? Is it the same simulation at different times? Or is it with different number of collocation points?

Sec. 3.1: “We optimize the simulation further by excluding vortex particles from the simulation when their vorticity falls below a certain threshold.” Are these particles turned back “on” later? Are they permanently removed from the simulation?

After Eq. 9, “x + h/2 w/w” Is this a typo? What is the results of vector w divided by itself? Was a normalization intended and the norm is missing at the denominator? The same for “x - h/2 w/w”.

What is the type of CPU / GPU used for the computation times reported in Table 1?

Most of the timings reported in Table 1 are above 1 ms, but the paper often insists on the idea that most simulations take less than 1 ms. Feels a bit misleading.

Typos:
* Naming convention issue: tracker vs tracer particles.
* In video, “vorticiy”
* Missing hyphen in “high resolution geometric models”, “Particle based physics”, “grid based fluid solvers”
* Spacing issue in “applications. By evolving”
* “The continuous velocity fields generated by the vortex particles allows” (should be allow). The same for “GPU-optimized VDB volumes by [21] allows”
* Framework instead of “frame-work”
* Incompressible instead of “In-compressible”
* Incorrect paragraph change (and related indentation) at “where p is the pressure,”
* “a superimposed collision fields” plural/singular inconsistency
* “physically accuracy” physical accuracy?
* “and k is the circulation of the segment. Defined by” Should be “segment, defined by”?
* Fig. 4, “Additional vortex particles (left) are advected” Should be (right)?
* “showing the interacting” interaction?
* Too much white space between Fig. 6 and its caption?

---

### Official Review · Reviewer_F9Vt · 2022-04-14
**A nice paper, but requires a little work to make it readable and reproducable.**

**Rating:** 9
**Confidence:** 4

**Review:**


Your sentence "a real-time capable fluid solver is not necessarily efficient enough for a game but should be able to create appealing fluid simulations in a fraction of a millisecond" doesn't make a lot of sense here; especially coupled with the previous part of the sentence "The time allocated [for] physics update[s]", values would be useful here (same with a "fraction of a millisecond").

You use the term "vortex shredding" this is a rarely used term and it is difficult to find any solid reference to this term, at first I thought it was "vortex shedding", but this doesn't not seem like a typo. The same for filament shredding - if you are using rarely used terms, it always helps to add an explanation for clarity.

Your use of direct references, such as "based on the work of [27]" makes hard reading, pretty much everyone familiar with Bridson or Stam's work will know them by their title, but I was continually having to refer to the references to determine what you were talking about. If you want a readable paper, then don't do this. Also, I think there are more seminal works for SPH than those listed.

It is not clear why the authors used a MaxQ version of the 2080 processor, this seems specifically for laptops and introduces some odd control elements make it difficult to compare (max q being developed to balance processing power and heat production - i.e. we are not sure when the limitations are kicking in).

Frames per second become meaningless at these speeds, what is more important is the amount of time spent and the percentage of the GPU used (both processor and memory).

It is not clear why 2.74 x 10^6 particles were used as a comparison to 1 x 10^6 particles. A whole number would have been more practical. It would have been useful to push the limits of the GPU to plot and determine whether there was an scaling issue (and where it occurred)

In the beginning, the authors' discuss the fact their method allows for adaptive sampling, but it's not really discussed again, it would be good to go into this in more detail in the method section, and show this is the case in the results.

In seems that while speed is perhaps not the issue, the other question is the effectiveness of using such methods - in several shots it's not really clear the character has any effect on the fluid/flame. In addition, flames are low viscosity and therefore reaction is low, to show this method more clearly a higher density fluid would have been better (such as water or oil).

It is not clear what is computational cost of the voxelisation process for rendering purposes; the time take should also be compared.

Why are you using an Adams-Bashforth integration method, and then comparing to Explicit Euler method? It seems there are other considerations (other methods) and would be useful to know why you picked this one (as opposed to RK integration for example)

Overall a quite nice method, difficult to know if the speed claims are credible, but they are within reason. The authors would do well to consider the above points, especially when using direct references - others may do it, but reading the paper is just awful and makes for hard work. Some more explanations here and there would also be useful.

---

### Decision · Program_Chairs · 2022-04-17

Accept